# The Relationship between Primary Energy Consumption and Real Gross Domestic Product: Evidence from Major Asian Countries

Wen-Chi Liu 

Department of Finance, Da-Yeh University, University Rd., Dacun, Changhua 51591, Taiwan;
vincent8@mail.dyu.edu.tw

**Abstract:** This study examines the relationship between primary energy consumption (PEC) and real gross domestic product (real GDP) in the top four major energy consumers in Asia, namely, China, India, Japan, and South Korea. The study period is from 1982–2018, covering 37 years of data after the second oil crisis (1979–1981). Bootstrap panel Granger causality method is applied to examine the causal relationship between PEC and real GDP. This method is capable of controlling cross-sectional dimension and cross-country heterogeneity. In addition, few studies investigate the relevance of real GDP to energy consumption, although real GDP adjusted by inflation provides an accurate picture of a country's economic situation. Our results contribute to existing literature in the field of PEC and real GDP. Through rigorous empirical research, we derive the main conclusion as follows. The real GDP and PEC of the top four energy consumers in Asia seem to be affected by the burst of the speculative Internet bubble from 2000–2001. Therefore, this study divides the research period into three periods: 1982–2018, 1982–2001, and 2002–2018. During the 1982–2018 period, an independent causal relationship is observed between real GDP and PEC for all four countries, thus supporting the neutrality hypothesis. During the 1982–2001 period, a unidirectional causal relationship running from PEC to real GDP is observed, thus supporting the energy growth hypothesis. Moreover, the coefficient is significantly negative in India; that is, PEC constrains economic development. Thus, the Indian government should reform its energy efficiency and consumption technologies to reduce energy waste. During the 2002–2018 period, an independent causal relationship is observed between real GDP and energy consumption for all four countries, thus supporting the neutrality hypothesis. This study then changes real GDP into nominal GDP and finds a unidirectional causal relationship running from PEC to nominal GDP in South Korea, thus supporting the growth hypothesis. A unidirectional causal relationship is also observed running from nominal GDP to PEC in India, thus supporting the energy conservation hypothesis. As mentioned above, we find that the relationship between PEC and real GDP adjusted by the GDP deflator is weaker than that between PEC and nominal GDP. Nominal GDP strengthens its relationship with PEC through the effect of prices for all the goods and services produced in an economy.**JEL Classification:** Q43; O47

**Keywords:** primary energy consumption; real gross domestic product; bootstrap panel granger causality test

## 1. Introduction

The British Petroleum Global Energy Statistical Yearbook 2019 reported that the growth in 2018 was remarkably strong because primary energy consumption (PEC) increased by 2.9% from 2017. In comparison, the average annual growth rate in the past decade was only 1.5%. Primary energy, also called natural energy, includes oil, natural gas, coal, nuclear power, hydropower, and renewable

energy power. It is divided into renewable and non-renewable energy. The former refers to natural energy that can be repeatedly generated, while the latter includes fossil and nuclear fuels that cannot be reused.

The level of primary energy prices deeply affects the profitability of various industries. Decreases in such prices are good for the economic growth (EG) of countries that rely on energy imports and can increase profits for manufacturers. The country's real gross domestic product (real GDP) will also increase, and the currency will tend to appreciate. However, increases in energy prices will likewise increase the production costs of most industries, negatively affecting the country's real GDP and causing the currency to depreciate.

Figure 1 shows the PEC (in million metric tons of oil equivalent, Mtoe) from 2015–2018, and Table 1 shows the data. The figure depicts that excluding natural gas, oil, coal, hydroelectricity, nuclear energy, and renewables among the primary energies show an upward trend. Table 1 reveals that oil was the most consumed primary energy source worldwide during the 2015–2018 period.

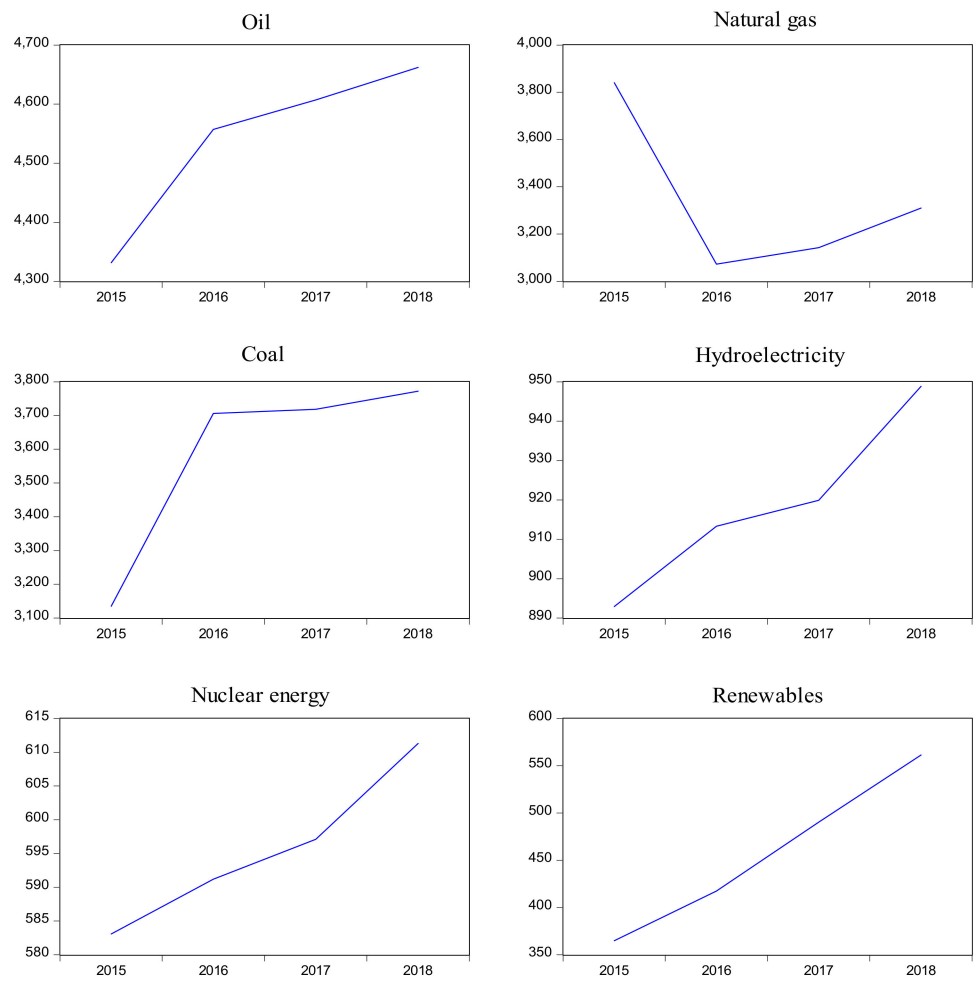

**Figure 1.** Trends of primary energy consumption (PEC) from 2015–2018 (in Mtoe).

**Table 1.** Data on PEC from 2015–2018 (in Mtoe).

| Year | Oil | Natural Gas | Coal | Hydroelectricity | Nuclear Energy | Renewables |
|------|------|-------------|--------|------------------|----------------|------------|
| 2015 | 4331.3 | 3839.9 | 3135.2 | 892.9 | 583.1 | 364.9 |
| 2016 | 4557.3 | 3073.2 | 3706 | 913.3 | 591.2 | 417.4 |
| 2017 | 4607 | 3141.9 | 3718.4 | 919.9 | 597.1 | 490.2 |
| 2018 | 4662 | 3309.4 | 3772.1 | 948.8 | 611.3 | 561.3 |

The price of New York's light crude oil was 144.53 USD/per barrel on 7 July 2008. After the global financial tsunami at the end of 2008, the price dropped to as low as 28.21 USD/per barrel on 19 January 2016, a decrease of up to 80.48%. Most Asian countries are oil-dependent countries that benefit when the price of oil imports decreases. For example, for China, consumption ability and manufacturing profits will increase because of lower oil prices. For Japan, reduced oil prices will positively affect its EG, and the export industry will be more competitive. For India, its high inflation rate and substantial current account deficits will be alleviated. Overall, the decline in oil prices has benefited Asian countries, such as Japan, South Korea, China, and India, helping them to engage in public construction and infrastructure, curb inflation, and increase GDP. Economists generally believe that energy consumption is related to economic activities and plays an important role in economic development. Thus, the relationship between energy consumption and GDP has received increased attention.

The burst of the speculative Internet bubble from 2000–2001 and the September 11 attacks in 2001 led to an important structural change for the world economy. The stock prices of technology-related and emerging Internet-related companies in the stock markets of North America, Europe, and Asia rose rapidly from 1995–2001. On the contrary, the global information technology industry severely declined in 2001, and the prosperity of the three major economies of the United States, Japan, and Europe fell in sync. Together with the September 11 attacks in 2011, this decline also affected the US economy and the world economy plummeted. The overall slowdown of world EG in 2001 was mainly due to the rare simultaneous decline of the three major economies, which affected the exports and EG of developing countries. The economies of Asian countries and the global information industry boom are highly interconnected. The burst of the speculative Internet bubble in the United States, through trade transmission, immediately and severely dampened the export of East Asian countries. Therefore, an important structural change occurred in 2002. This study divides the research period into three periods, namely, 1982–2018, 1982–2001, and 2002–2018 to obtain robust results.

This study first uses the bootstrap causality test to determine the relationship between energy consumption and real GDP for the top four major energy consumers in Asia, namely, China, India, Japan, and South Korea. This test can effectively overcome cross-sectional correlation and heterogeneity problems. Moreover, few studies investigate the relevance of real GDP to energy consumption, even though real GDP adjusted by inflation provides an accurate picture of a country's economic situation. The results of this study will contribute to the academic and practical fields.

## 2. Literature Review

For the past few decades, scholars have been examining the relationship between GDP (EG) and energy consumption. Kraft and Kraft [1] were the first to research the relationship between energy consumption and GNP, and some scholars continued to use different methods and select different countries to explore this relationship. Ozturk [2], Payne [3], and Al-mulali et al. [4] demonstrated four hypotheses of energy consumption and GDP (EG).

**Hypothesis 1 (H1):** *The growth hypothesis proposes that energy consumption Granger affects GDP (EG).*

**Hypothesis 2 (H2):** *The energy conservation hypothesis proposes that GDP (EG) Granger affects energy consumption.*

**Hypothesis 3 (H3):** *The feedback hypothesis regards the bidirectional causality between GDP (EG) and energy consumption.*

**Hypothesis 4 (H4):** *The neutrality hypothesis posits that GDP (EG) and energy consumption are independent.*

Mathur et al. [5] suggested that for developing and transition economies, per capita energy consumption has an adverse effect on EG per capita. However, for developed countries, per capita

energy consumption has a positive impact on EG per capita. The latter may be related to the notion that developed countries use more alternative energies than developing countries. The following studies support one of the four cited hypotheses on energy consumption and GDP (EG).

Narayan and Smyth [6], Ang [7], Chontanawat et al. [8], and Sari and Soytas [9] supported the growth hypothesis and considered that a one-way causal relationship runs from energy consumption to EG. Azam [10] revealed that energy has a significant positive impact on EG in developing Asian economies. Chiou-Wei et al. [11] revealed that energy consumption affects EG for Taiwan, Hong Kong, Malaysia, and Indonesia. Meidani and Zabihi [12] also showed that a strong Granger causality runs from energy consumption to GDP in the Iranian industry sector. Magazzino [13] found that a short-run one-way causality runs from energy consumption to GDP in Italy. Mahalingam and Orman [14] indicated that energy consumption Granger affects GDP in the Rocky Mountain region of the United States, thus supporting the growth hypothesis. These findings mean that energy consumption plays an important role in EG. If a positive causal relationship is observed between energy consumption and GDP (EG), then energy consumption can positively affect GDP (EG), that is, the GDP (EG) increases as energy consumption increases. On the contrary, if a negative causal relationship exists between energy consumption and GDP (EG), then energy consumption can negatively affect GDP (EG). Therefore, energy constrains GDP (EG), such that the energy supply will have an adverse effect on GDP (EG).

Kuo et al. [15], Lee and Chien [16], Apergis and Payne [17], and Lee [18] proposed the energy conservation hypothesis, which states that a one-way causality runs from GDP (EG) to energy consumption. Chen et al. [19] revealed that a unidirectional short-run causality runs from EG to electricity consumption in 10 Asian countries. Chiou-Wei et al. [11] revealed that a unidirectional causality runs from EG to energy consumption in the Philippines and Singapore. Magazzino [20] also indicated that GDP affects energy consumption. Mahalingam and Orman [14] indicated that in the Southwest of the United States, GDP Granger affects energy consumption, thus supporting the conservation hypothesis. Reducing energy consumption or increasing energy efficiency to reduce carbon dioxide emissions and the greenhouse effect will not hinder economic development and growth. Governments can adopt a strict energy consumption policy and facilitate the promotion and implementation of energy environmental protection activities. This action is beneficial for the global environment, and the country's GDP will not decline. Evidence from Lebanon, Fakih, and Marrouch [21] also supports this viewpoint.

Zhang and Broadstock [22], Arora and Shuping [23], Gross [24], Eggoh and Rault [25], Zhixin and Xin [26], Belloumi [27], Chima and Freed [28], and Ghali and El-Sakka [29] supported the feedback hypothesis and believed that a two-way causal relationship exists between EG and energy consumption. Magazzino [13] revealed a long-run bidirectional causal relationship between energy consumption and GDP in Italy. Salmanzadeh-Meydani and Fatemi Ghomi [30] showed a bidirectional long-run causality between electricity consumption and EG in Iran. Ghali and El-Sakka [29] argued that a two-way Granger causality exists between EG and energy consumption in Canada. Jaiyesimi et al. [31] also revealed the presence of two-way causality between energy consumption and GDP in the Organization for Economic Cooperation and Development. They found that reducing energy consumption has a negative effect on GDP, which constrains EG, and vice versa.

Fang and Wolski [32] supported the neutral hypothesis that energy consumption and EG are independent in China. In addition, Chiou-Wei et al. [11] supported the neutrality hypothesis for the United States, Thailand, and South Korea.

Different conclusions in the current literature of the relationship between energy consumption and GDP (EG) were obtained, yet no literature used the bootstrap panel Granger causality test. In addition, even though real GDP provides an accurate picture of a country's economic situation, few studies investigate the relevance of real GDP to energy consumption. Hence, the current study uses bootstrap panel Granger causality method and real GDP data to investigate the top four major energy consumers

in Asia, namely, China, India, Japan, and South Korea. The findings of this study will contribute to the academic and practical fields.

## 3. Data

The second oil crisis began at the end of 1978 during the Iranian "Islamic Revolution", during which time oil exports were interrupted. The crisis lasted more than two years, and ended in early 1981. Thus, this study period covered the 37 years after the second oil crisis, from 1982–2018, and focused on the top four primary energy consumers in Asia, namely, China, India, Japan, and South Korea. From the Global Energy Statistical Yearbook 2019, the energy consumption of these four countries in 2018 is 3273.47 Mtoe in China (first in the world), 809.15 Mtoe in India (third in the world), 454.14 Mtoe in Japan (fifth in the world), and 301.02 Mtoe in South Korea (eighth in the world).

Oil, natural gas, coal, nuclear power, hydropower, and renewable energy power belong to primary energy. PEC data were taken from the British Petroleum Global Energy Statistical Yearbook 2019, while the real GDP data (in constant 2010 million USD) and nominal GDP data (in current million USD) are taken from the World Bank. Real GDP provides an accurate picture of a country's economic situation because it can be easily compared to past data adjusted by inflation. Therefore, whether a country's situation is better or worse year by year can be inferred.

## 4. Methodology

In this study, the bootstrap panel Granger causality test is used as the research method. Before conducting the test, cross-sectional dependence and slope homogeneity tests must be performed. The bootstrap panel Granger causality test can obtain more robust results when the data possess cross-sectional dependence and slope heterogeneity problems. The following is the description for three tests: Cross-sectional dependence, slope homogeneity, and bootstrap panel Granger causality.

### 4.1. Cross-Sectional Dependence Test

The Lagrangian multiplier (LM represented below) of Breusch and Pagan [33], one of the cross-sectional dependence tests, is conducted to verify the cross-sectional dependence. The null hypothesis of without cross-section dependence is shown as $H_0 : \text{Cov}(u_{it}, u_{jt}) = 0$, and the alternative hypothesis of cross-sectional dependence is shown as $H_1 : \text{Cov}(u_{it}, u_{jt}) \neq 0$. Testing the null hypothesis, Breusch and Pagan [33] set the LM test as follows (Equation (1)):

$$\text{LM} = T \sum_{i=1}^{N-1} \sum_{j=i+1}^{N} \hat{p}_{ij}^2, \tag{1}$$

where $\hat{p}_{ij}$ is the sample estimation for the pairwise residual correlation. Under this null hypothesis, the LM statistic has asymmetric chi-square test with degrees of freedom: $N(N-1)/2$. For large samples of $T \to \infty$ and $N \to \infty$, Pesaran [34] recommended a shortened version of the $\text{CD}_{\text{lm}}$ test as follows (Equation (2)):

$$\text{CD}_{\text{lm}} = \left( \frac{1}{N(N-1)} \right)^{1/2} \sum_{i=1}^{N-1} \sum_{J=i+1}^{N} (T\hat{p}_{ij}^2 - 1). \tag{2}$$

Under this null hypothesis, the $\text{CD}_{\text{lm}}$ test covers the standard normal distribution. When $N$ is large and $T$ is small, the $\text{CD}_{\text{lm}}$ test faces substantial error distortion. For the sample of $T \to \infty$ and $N \to \infty$, Pesaran [34] set a more general valid cross-section dependence test. The following is the CDtest (Equation (3)):

$$\text{CD} = \sqrt{\left( \frac{2T}{N(N-1)} \right)} \left( \sum_{i=1}^{N-1} \sum_{J=i+1}^{N} \hat{p}_{ij} \right). \tag{3}$$

The CD test has an asymptotic standard normal distribution under the null hypothesis. Pesaran [34] pointed out that the CD test is also powerful for heterogeneous dynamic models that have fixed $T$ and $N$ with absolute mean zero values and contain multiple variable slope coefficients and/or a heterogeneous dynamic model of error variables. The CD test cannot be verified when the overall average pairwise correlation is zero and the potential individual average pairwise correlation is non-zero. Pesaran et al. [35] set the error-corrected version of the test using the absolute means and variables of LM statistics. The bias-adjusted test is as follows (Equation (4)):

$$
\text{LM}_{\text{adj}} = \sqrt{\left(\frac{2T}{N(N-1)}\right)} \sum_{i=1}^{N-1} \sum_{j=i+1}^{N} \hat{p}_{ij} \frac{(T-k)\hat{p}_{ij}^2 - \mu T_{ij}}{\sqrt{v_{T_{ij}}^2}},
\tag{4}
$$

where $\mu T_{ij}$ and $v_{T_{ij}}^2$ set by Pesaran et al. [35] are the absolute mean and variables of $(T-k)\hat{p}_{ij}^2$, respectively, and $k$ represents the number of independent variables. Under the null hypothesis of $T \to \infty$ and $N \to \infty$, the asymptotic distribution of the $\text{LM}_{\text{adj}}$ test is the standard normal.

### 4.2. Slope Homogeneity Test

The Wald statistic is used for slope homogeneity test and is effective for cases with small cross-sections and large time series, the independent variables being absolute exogenous, and the error variables being homogeneous.

Using the Wald statistic, Swamy [36] test is applicable to the sample data that $N$ is relatively smaller than $T$. However, Pesaran and Yamagata [37] employed a standardized version of Swamy's test (the $\widetilde{\Delta}$ test) to verify the slope homogeneity in large samples. When the error terms are normally distributed and $(N, T) \to \infty$, the relative dilatability of $N$ and $T$ has no restriction, and the test is valid. In the $\widetilde{\Delta}$ test method, the first step is to calculate an adjusted version of the following Swamy test (Equation (5)):

$$
\widetilde{S} = \sum_{i=1}^{N} \left(\widehat{\beta}_i - \widetilde{\beta}_{\text{WFE}}\right)' \frac{x'_i M_t x_i}{\widetilde{\sigma}_i^2} \left(\widehat{\beta}_i - \widetilde{\beta}_{\text{WFE}}\right).
\tag{5}
$$

where $\widehat{\beta}_i$ is the common ordinary least squares, $\widetilde{\beta}_{\text{WFE}}$ is the common valuation for weighted fixed effects, $M_t$ is the unit matrix, and $\widetilde{\sigma}_i^2$ is the valuation of $\sigma_i^2$. Standard deviation statistics are as follows (Equation (6)):

$$
\widetilde{\Delta} = \sqrt{N}\left(\frac{N^{-1}\widetilde{S} - k}{\sqrt{2k}}\right),
\tag{6}
$$

where $k$ is the number of independent variables. When the null hypothesis under the condition of $(N, T) \to \infty$, as long as $\sqrt{N}/T \to \infty$ and the error term is a normal distribution, the $\widetilde{\Delta}$ test will present an asymptotic standard normal distribution. The following error correction version can improve the small sample characteristics of the test (Equation (7)):

$$
\widetilde{\Delta}_{\text{adj}} = \sqrt{N}\left(\frac{N^{-1}\widetilde{S} - E\left(\widetilde{Z}_{it}\right)}{\sqrt{\text{var}\left(\widetilde{Z}_{it}\right)}}\right),
\tag{7}
$$

where $E\left(\widetilde{Z}_{it}\right) = k$ and $\text{var}\left(\widetilde{Z}_{it}\right) = 2k(T-k-1)/T + 1$.

### 4.3. Bootstrap Panel Granger Causality Test

The concept of the bootstrap Granger panel causality test is to use the preceding value of a variable to foresee the future value of another variable. This method will supply additional information and reliable statistical results than the time series method. This study uses the bootstrap panel Granger

causality test proposed by Kónya [38] to determine the causal relationship between primary energy consumption and real GDP. Kónya [38] accentuated that the bootstrap panel Granger causality test can highlight the unit root (nonstationary) and cointegration characteristics of variables, that is, the verification process does not need the pre-tests of unit root and cointegration. Given that these variables do not need to consider the time series characteristics, we can use its critical value instead.

The first step of Kónya's [38] bootstrap panel causality test is to perform the Wald test by the seemingly unrelated regression (SUR) model to impose a zero-causal relationship. Then, the critical value of sampling is obtained. In this bootstrap panel causality test, joint assumptions are not required for all sample countries due to the country-specific sampling critical value using the Wald test.

The causal analysis system for this sample contains two sets of calculation formulas (Equation (8)):

$$
\begin{aligned}
y_{1,t} &= \alpha_{1,1} + \sum_{i=1}^{ly_1} \beta_{1,1,i} y_{1,t-i} + \sum_{i=1}^{lx_1} \delta_{1,1,i} x_{k,1,t-i} + \varepsilon_{1,1,t}, \\
y_{2,t} &= \alpha_{1,2} + \sum_{i=1}^{ly_1} \beta_{1,2,i} y_{2,t-i} + \sum_{i=1}^{lx_1} \delta_{1,2,i} x_{k,2,t-i} + \varepsilon_{1,2,t}, \\
y_{N,t} &= \alpha_{1,N} + \sum_{i=1}^{ly_1} \beta_{1,N,i} y_{N,t-i} + \sum_{i=1}^{lx_1} \delta_{1,N,i} x_{k,N,t-i} + \varepsilon_{1,N,t}
\end{aligned}
\tag{8}
$$

And (Equation (9)):

$$
\begin{aligned}
x_{k,1,t} &= \alpha_{2,1} + \sum_{i=1}^{lx_2} \delta_{2,1,i} x_{k,1,t-i} + \sum_{i=1}^{ly_2} \beta_{2,1,i} y_{1,t-i} + \varepsilon_{2,1,t}, \\
x_{k,2,t} &= \alpha_{2,2} + \sum_{i=1}^{lx_2} \delta_{2,2,i} x_{k,2,t-i} + \sum_{i=1}^{ly_2} \beta_{2,2,i} y_{2,t-i} + \varepsilon_{2,2,t}, \\
x_{k,N,t} &= \alpha_{2,N} + \sum_{i=1}^{lx_2} \delta_{2,N,i} x_{k,N,t-i} + \sum_{i=1}^{ly_2} \beta_{2,N,i} y_{N,t-i} + \varepsilon_{2,N,t}
\end{aligned}
\tag{9}
$$

where $N$ represents the number of country samples ($i = 1, \dots, N$), $t$ represents the period, and $l$ represents the number of lag periods. In Equations (8) and (9), each formula contains different predetermined variables. When an error term exists, a cross-linking effect may occur. These calculated formulas belong to the SUR theory, where x is primary energy consumption and y is real GDP (EG). The Granger causality test can contain the following four definitions:

*x->y*: Unidirectional Granger causality supports the growth hypothesis, given that not all $\delta_{1,i}$ are zero, but all $\beta_{2,i}$ are zero.

*y->x*: Unidirectional Granger causality supports the energy conservation hypothesis, given that all $\delta_{1,i}$ are zero, but not all $\beta_{2,i}$ are zero.

*x<->y*: Bidirectional Granger causality supports the feedback hypothesis, given that $\delta_{1,i}$ and $\beta_{2,i}$ are not zero.

*x<≠>y*: No Granger causality supports the neutrality hypothesis, given that $\delta_{1,i}$ and $\beta_{2,i}$ are zero.

Prior to empirical studies, the optimal number of lag periods must be determined. The result of the bootstrap Granger panel causality test may be sensitive to the number of lag periods. Hence, the optimal number of lag periods must be verified to achieve robustness. Kónya [38] noted that the maximum number of lag periods can vary depending on the variables, but the calculation formulas must be the same. The Schwarz information criterion is used to obtain the optimum lag period.

## 5. Empirical Results and Policy Implications

The research period of this study is from 1982–2018, covering 37 years of data. The objective is to investigate the relationship between PEC (Mtoe) and real GDP (constant 2010 million USD) in four Asian countries. This study utilizes the bootstrap causality test to effectively overcome cross-sectional correlation and heterogeneity problems and determine the relationship between PEC and real GDP of China, India, Japan, and South Korea.

Tables 2 and 3 show the descriptive statistics of PEC and real GDP, respectively. With regard to the mean statistic, China and Japan have the highest energy consumption and real GDP, respectively. In addition, the Jarque–Bera statistics indicate that all four countries have a near-normal distribution in PEC. With regard to the real GDP, South Korea has a near-normal distribution, in contrast to China, India, and Japan.

**Table 2.** Descriptive statistics of PEC for the four countries.

| Statistic | China | India | Japan | South Korea |
|---|---|---|---|---|
| Mean | 1511 | 365 | 466 | 179 |
| Median | 1011 | 318 | 473 | 194 |
| Maximum | 3273 | 809 | 531 | 301 |
| Minimum | 429 | 113 | 341 | 41 |
| Std. Dev. | 956 | 202 | 55 | 85 |
| Skewness | 0.582 | 0.644 | −0.801 | −0.233 |
| Kurtosis | 1.769 | 2.244 | 2.635 | 1.721 |
| Jarque-Bera | 4.423 | 3.436 | 4.164 | 2.857 |
| Probability | 0.110 | 0.179 | 0.125 | 0.240 |
| Observations | 37 | 37 | 37 | 37 |

**Table 3.** Descriptive statistics of real gross domestic product (GDP) for the four countries.

| Statistic | China | India | Japan | South Korea |
|---|---|---|---|---|
| Mean | 3,520,935 | 1,107,718 | 5,126,468 | 729,411 |
| Median | 2,232,146 | 873,357 | 5,348,935 | 710,035 |
| Maximum | 10,797,222 | 2,841,580 | 6,189,778 | 1,381,860 |
| Minimum | 390,229 | 324,235 | 3,250,668 | 163,676 |
| Std. Dev. | 3,156,100 | 721,372 | 817,033 | 380,629 |
| Skewness | 0.920 | 0.889 | −0.892 | 0.127 |
| Kurtosis | 2.547 | 2.684 | 2.787 | 1.729 |
| Jarque-Bera | 5.537 | 5.023 | 4.972 | 2.591 |
| Probability | 0.063 | 0.081 | 0.083 | 0.274 |
| Observations | 37 | 37 | 37 | 37 |

Figure 2 shows that the trends of real GDP (constant 2010 million USD) and PEC (Mtoe) are remarkably consistent, increasing from 1982–2018 in China, India, and South Korea. In addition, Japan's real GDP shows an upward trend from 1982–2018, while its PEC shows an upward trend from 1982–2005, but a slightly downward trend from 2006–2018.

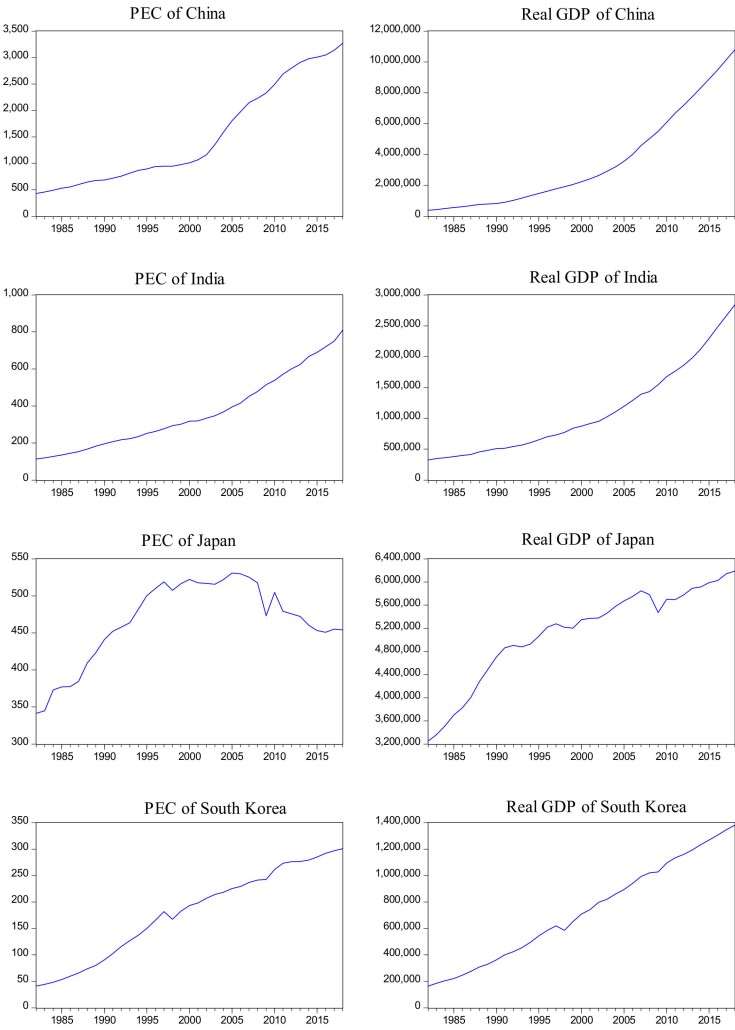

**Figure 2.** Real GDP and PEC for the four countries from 1982–2018.

From Table 4, the results of the trend tests of eight variables show that all the coefficients of trend are positive and significant at the 1% probability level. When the regression model has a trend effect, the coefficient value is generally large, thereby improving the overall significance (F value) and rejecting the null hypothesis. Figure 2 and Table 4 show that the PEC and real GDP of the four countries are generally affected by a certain degree of trend. Therefore, the subsequent empirical process will remove the trend influence of all eight variables so that the independent variables can properly explain the dependent variables.

**Table 4.** Trend tests of eight variables.

| Variables | Intercept | Trend |
|---|---|---|
| China's PEC | 6.038918 *** | 0.059896 *** |
| China's real GDP | 12.94175 *** | 0.093533 *** |
| India's PEC | 4.790020 *** | 0.053110 *** |
| India's real GDP | 12.63306 *** | 0.060058 *** |
| Japan's PEC | 6.014741 *** | 0.006840 *** |
| Japan's real GDP | 15.16895 *** | 0.014828 *** |
| South Korea's PEC | 4.064293 *** | 0.054000 *** |
| South Korea's real GDP | 12.30509 *** | 0.057040 *** |

Note: *** indicates significance at the 1% probability level.

Kónya [38] emphasized that the bootstrap panel Granger causality test can highlight the unit root (nonstationary) and cointegration characteristics of variables; that is, the verification process does not need the pre-tests of unit root and cointegration. Nevertheless, the pre-tests of cross-sectional dependence and slope homogeneity are necessary prior to conducting the bootstrap panel Granger causality test.

The results of four cross-sectional related tests in Table 5 and two homogeneity tests in Table 6 show that cross-sectional correlation and heterogeneity problems exist between energy consumption and real GDP for all four countries. Thus, this study will use the bootstrap panel Granger causality test to effectively overcome the cross-sectional correlation and heterogeneity problems.

**Table 5.** Cross-sectional dependence test of PEC and real GDP.

| Cross-Sectional Dependence Test | Stat. | P Value |
| --- | --- | --- |
| Breusch and Pagan (1980) $CD_{BP}$ | 13.957 ** | 0.0301 |
| Pesaran [34] $CD_{LM}$ | 2.297 ** | 0.0216 |
| Pesaran [34] $CD$ | 2.404 ** | 0.0162 |
| Pesaran, Ullah, and Yamagata [35] $LM_{adj}$ | 139.4741 *** | 0.0000 |

Note: ***, and ** indicate significance at the 1% and 5% probability levels, respectively.

**Table 6.** Slope homogeneity test for PEC and real GDP.

| Slope Homogeneity Test | Stat. | P Value |
| --- | --- | --- |
| Swamy [36] $\hat{S}$ | 8.7472 ** | 0.0328 |
| Pesaran and Yamagata [37] $\widetilde{\Delta}$ | 1.6784 ** | 0.0466 |

Note: ** indicates significance at the 5% probability level.

Prior to the bootstrap Granger causality test, this study performs the VAR optimum lag period test using the Schwarz information criterion. The results show the following: Lag 2 (China), lag 1 (India), lag 2 (Japan), and lag 2 (South Korea). Therefore, the number of the maximum optimal lag period is 2, which will be used in the bootstrap panel Granger causality test.

Figure 3 illustrates that the real GDP and PEC of the four countries seem to be affected by the burst of the speculative Internet bubble from 2000–2001 and the September 11 attacks in 2001, thus causing structural change. Therefore, this study divides the research period into three periods, namely, 1982–2018, 1982–2001, and 2002–2018, to obtain more robust results. The shaded area is the 2002–2018 period.

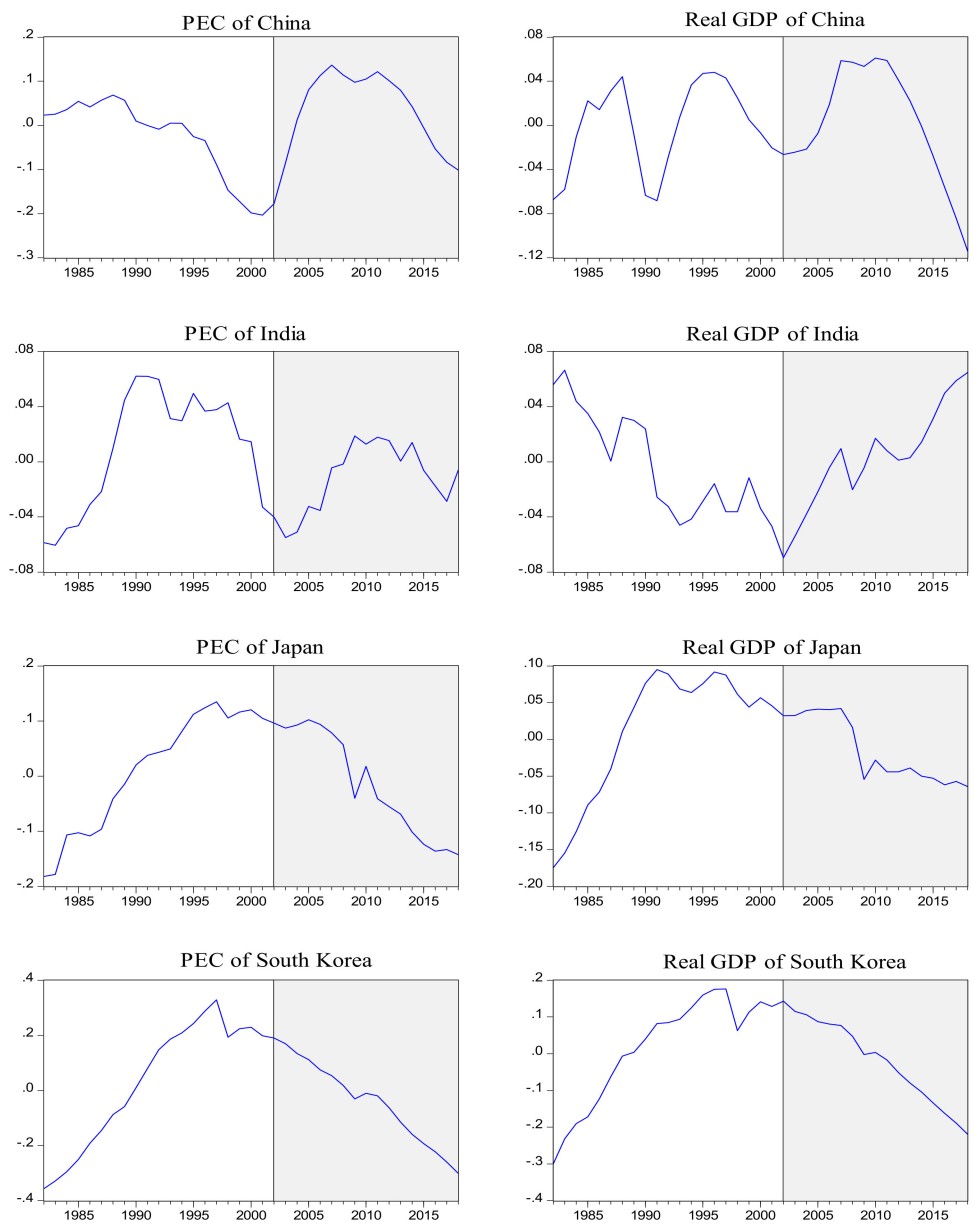

**Figure 3.** Real GDP and PEC for the four countries from 1982– 2018 (detrend).

The bootstrap panel Granger causality test is capable of controlling cross-sectional dimension and cross-country heterogeneity and obtains robust results. Tables 7 and 8 show the results of the bootstrap panel Granger causality test for the 1982–2018 period, in which critical values are obtained from 50,000 bootstrap replications. From the tables, an independent causal relationship is observed between real GDP and PEC for all four countries, thus supporting the neutrality hypothesis.

**Table 7.** Bootstrap panel Granger causality test from PEC to real GDP (1982–2018).

| Country | Wald Statistics (Chi-Square) | Bootstrap Critical Value | | |
|---|---|---|---|---|
| | | 10% | 5% | 1% |
| China | 1.30906 | 9.36029 | 12.41730 | 19.87257 |
| India | 2.84448 | 8.39919 | 11.14984 | 18.34286 |
| Japan | 0.64922 | 9.99257 | 13.02899 | 20.30982 |
| South Korea | 0.22646 | 11.13418 | 14.61762 | 23.25567 |

Note: The critical values are obtained from 50,000 bootstrap replications.

**Table 8.** Bootstrap panel Granger causality test from real GDP to PEC (1982–2018).

| Country | Wald Statistics (Chi-Square) | Bootstrap Critical Value | | |
|---|---|---|---|---|
| | | 10% | 5% | 1% |
| China | 1.80372 | 7.23608 | 9.64457 | 15.51614 |
| India | 1.00317 | 8.11879 | 10.75914 | 17.41479 |
| Japan | 0.48558 | 9.92862 | 12.95030 | 20.25928 |
| South Korea | 1.19111 | 10.56256 | 13.83535 | 22.47874 |

Note: The critical values are obtained from 50,000 bootstrap replications.

Tables 9 and 10 show the results of the bootstrap panel Granger causality test for the 1982–2001 period, in which critical values are obtained from 50,000 bootstrap replications. Table 11 also shows the result of the coefficient analysis of the independent variable, namely, PEC. From Tables 9 and 10, a unidirectional causal relationship is observed running from PEC to real GDP in India, thus supporting the energy growth hypothesis that PEC has a significant impact on real GDP. Furthermore, from the coefficient analysis results of the independent variables, PEC in Table 11, we can see that the sum of the coefficients of lags 1 and 2 is –0.39109 for India. This finding means that one Mtoe increase in PEC will cause real GDP to decrease 0.391089 million USD in the following two years. Specifically, PEC constrains economic development and has an adverse effect on the economy. This situation may be caused by poor energy efficiency, poor energy consumption technology, and energy waste. Thus, the Indian government should reform its energy efficiency and consumption technologies to reduce energy waste.

**Table 9.** Bootstrap panel Granger causality test from PEC to real GDP (1982–2001).

| Country | Wald Statistics (Chi-Square) | Bootstrap Critical Value | | |
|---|---|---|---|---|
| | | 10% | 5% | 1% |
| China | 5.55825 | 11.89077 | 16.44711 | 28.91109 |
| India | 15.73870 * | 14.58854 | 19.85390 | 33.74493 |
| Japan | 4.69540 | 15.28616 | 19.96515 | 32.34416 |
| South Korea | 1.78485 | 16.35465 | 25.40176 | 66.44516 |

Note: 1. * indicates significance at the 10% probability level; 2. The critical values are obtained from 50,000 bootstrap replications.

**Table 10.** Bootstrap panel Granger causality test from real GDP to PEC (1982–2001).

| Country | Wald Statistics (Chi-Square) | Bootstrap Critical Value | | |
|---|---|---|---|---|
| | | 10% | 5% | 1% |
| China | 8.37381 | 10.88091 | 15.28721 | 27.39948 |
| India | 0.55319 | 11.36074 | 15.31303 | 26.73922 |
| Japan | 1.55375 | 16.35401 | 21.97498 | 38.17194 |
| South Korea | 5.23025 | 17.26062 | 27.47691 | 61.04502 |

Note: The critical values are obtained from 50,000 bootstrap replications.

**Table 11.** Results of coefficient analysis of the independent variable, PEC (1982–2001).

| Country | Coefficient Estimation of PEC | |
|---|---|---|
| | Lag 1 | Lag 2 |
| China | −0.501153 ** | 0.534917 ** |
| | (0.216022) | (0.246036) |
| | {−2.31992} | {2.17415} |
| India | 0.303933 | −0.695022 *** |
| | (0.194007) | (0.217534) |
| | {1.56661} | {−3.19500} |
| Japan | 0.254185 * | −0.280351 ** |
| | (0.132550) | (0.129445) |
| | {1.91765} | {−2.16579} |
| South Korea | −0.364065 | 0.119084 |
| | (0.342026) | (0.250588) |
| | {−1.06444} | {0.475218} |

Note: 1. ***, **, and * indicate significance at the 1%, 5%, and 10% probability levels, respectively; 2. Numbers in parentheses are the standard error, and those in curly brackets are the t value; 3. The critical value is taken from the Student's *t* test probability distribution table.

Tables 12 and 13 show the results of the bootstrap panel Granger causality test for the 2002–2018 period, in which critical values are obtained from 50,000 bootstrap replications. From Tables 12 and 13, an independent causal relationship is observed between real GDP and PEC for all four countries, thus supporting the neutrality hypothesis.

**Table 12.** Bootstrap panel Granger causality test from PEC to real GDP (2002–2018).

| Country | Wald Statistics (Chi-Square) | Bootstrap Critical Value | | |
|---|---|---|---|---|
| | | 10% | 5% | 1% |
| China | 4.79630 | 18.15488 | 25.15471 | 47.20276 |
| India | 1.05641 | 13.82703 | 19.49366 | 36.63676 |
| Japan | 3.04535 | 17.36349 | 24.99200 | 48.41681 |
| South Korea | 0.23664 | 16.35295 | 23.43130 | 44.32858 |

Note: The critical values are obtained from 50,000 bootstrap replications.

**Table 13.** Bootstrap panel causality test from real GDP to PEC (2002–2018).

| Country | Wald Statistics (Chi-Square) | Bootstrap Critical Value | | |
|---|---|---|---|---|
| | | 10% | 5% | 1% |
| China | 4.35962 | 21.03115 | 28.57598 | 49.66736 |
| India | 7.88699 | 12.33403 | 17.07066 | 31.28848 |
| Japan | 1.35612 | 17.00388 | 27.05964 | 64.29070 |
| South Korea | 6.32712 | 19.85489 | 27.03482 | 45.98591 |

Note: The critical values are obtained from 50,000 bootstrap replications.

For comparative purpose, this study also reveals the results of nominal GDP in Tables 14 and 15. In Table 14, a unidirectional causal relationship is observed running from PEC to nominal GDP in South Korea, thus supporting the growth hypothesis that PEC has a significant effect on nominal GDP. In Table 15, a unidirectional causal relationship is observed running from nominal GDP to PEC in India, thus supporting the energy conservation hypothesis that nominal GDP has a significant impact on PEC.

**Table 14.** Bootstrap panel causality test from PEC to nominal GDP (2002–2018).

| Country | Wald Statistics (Chi-Square) | Bootstrap Critical Value | | |
|---|---|---|---|---|
| | | 10% | 5% | 1% |
| China | 10.05593 | 19.07468 | 26.03238 | 48.00904 |
| India | 0.51926 | 14.35847 | 20.31498 | 38.90902 |
| Japan | 0.67267 | 11.92841 | 16.66543 | 31.03997 |
| South Korea | 11.72555 * | 10.58405 | 15.00677 | 28.40736 |

Note: 1. * indicates significance at the 10% probability level; 2. The critical values are obtained from 50,000 bootstrap replications.

**Table 15.** Bootstrap panel causality test from nominal GDP to PEC (2002–2018).

| Country | Wald Statistics (Chi-Square) | Bootstrap Critical Value | | |
|---|---|---|---|---|
| | | 10% | 5% | 1% |
| China | 0.13585 | 18.44549 | 25.03878 | 44.11636 |
| India | 14.53784* | 14.33181 | 20.28594 | 36.72123 |
| Japan | 1.35186 | 12.96341 | 18.56781 | 33.27691 |
| South Korea | 2.28376 | 10.91452 | 15.46182 | 26.65157 |

Note: 1. * indicates significance at the 10% probability level; 2. The critical values are obtained from 50,000 bootstrap replications.

As mentioned above, we find that the relationship between PEC and real GDP adjusted by the GDP deflator is weaker than that between PEC and nominal GDP. Nominal GDP strengthens its relationship with PEC through the effect of prices for all the goods and services produced in an economy.

## 6. Conclusions

This study mainly examines the causal relationship between the PEC and real GDP of the top four major energy consumers in Asia, namely, China, India, Japan, and South Korea. The study period is from 1982–2018, covering 37 years of data after the second oil crisis in 1979–1981. However, the PEC and real GDP of these four countries seem to be affected by the burst of the speculative Internet bubble from 2000 to 2001. Therefore, this study divides the research period into three periods, namely, 1982–2018, 1982-2001, and 2002–2018, to obtain more robust results.

During the 1982–2018 period, an independent causal relationship is observed between real GDP and PEC for all four countries, thus supporting the neutrality hypothesis. This finding is similar to those of Fang and Wolski [32] and Chiou-Wei et al. [11], thus supporting the neutrality hypothesis in China and in the United States, Thailand, and South Korea, respectively. During the 1982–2001 period, a unidirectional causal relationship is observed running from PEC to real GDP in India, thus supporting the energy growth hypothesis that PEC has a significant impact on real GDP. This finding is similar to those of Azam [10] and Chiou-Wei et al. [11], thus supporting the energy growth hypothesis in developing Asian economies and in Taiwan, Hong Kong, Malaysia, and Indonesia, respectively. Additionally, the sum of coefficients of lags 1 and 2 is –0.39109 for India. Specifically, one Mtoe increase in PEC will cause real GDP to decrease by 0.391089 million USD in the following two years. PEC constrains economic development, such that it will have an adverse effect on the economy in India. Thus, the Indian government should reform its energy efficiency and consumption technologies to reduce energy waste.

During the 2002–2018 period, an independent causal relationship is observed between real GDP and PEC for all four countries, thus supporting the neutrality hypothesis. This study also compares the results of real GDP with those of nominal GDP. A unidirectional causal relationship is observed running from PEC to nominal GDP in South Korea, thus supporting the growth hypothesis that energy consumption has a significant effect on nominal GDP. A unidirectional causal relationship is observed running from nominal GDP to PEC in India, thus supporting the energy conservation hypothesis that nominal GDP has a significant impact on energy consumption. As mentioned above, the relationship

between PEC and real GDP adjusted by the GDP deflator is weaker than that between PEC and nominal GDP. Nominal GDP strengthens its relationship with PEC through the effect of prices for all the goods and services produced in an economy. The innovative findings of this study will contribute to this line of research.

**Funding:** This research received no external funding.

**Conflicts of Interest:** The authors declare no conflict of interest.

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
