# Peer review of "The Relationship between Primary Energy Consumption and Real Gross Domestic Product: Evidence from Major Asian Countries"

_sustainability, doi:10.3390/su12062568_

Round 1
Reviewer 1 Report
The artcle has great interest, but it must be improved in several aspects to recommend the publication in this Journal.
The author does not adequately present the goal. For example, he makes many sentences in the introduction without applied studies that support the question, it is not clear the study process, or the reasons that justify the choice of sample both in entities and in period.
A deeper review of the relationship analyzed is necessary, the description of the variables and their methodological treatment must also be included beyond the tests listed.
It is not clear to me that "real GDP is measured in current units", if the price effect has not been taken into account, the study cannot be considered.
Finally, the article should be rewritten more scientifically, discussing results and drawing real conclusions.
Other minor issues are the treatment of acronyms and references, see the rules of the journal.
Author Response
Thanks for your comments and suggestions.
Responses:
- The introduction and conclusions have been rewritten in this study.
- This study has referred to the rules of this journal to update references.
- Attached is the full text.

Reviewer 2 Report
This study is nicely organized and the presented concepts (overall idea, literature review, data, methodology, results...).
However, I suggest that the authors try to investigate is there a structural break in the data. Graphs show that this might be the case for some countries.
Also, in addition to the methodology used in this paper, try doing the same thing with another econometric method in order to see if the results are comparable and/or which method definitively give (more) robust results. If you will be doing so, try using for example ARDL approach.
Also, the Conclusion part should be more detailed and longer than it is right now. The "policy recommendation" part should be precise containing several detailed recommendations. This is the purpose of such papers: determine the causal nexus and provide policies for responsible authorities.
Author Response
Thanks for your comments and suggestions.
Responses:
- From Figure 3, it can be seen that the GDP and primary energy consumption (PEC) of these four countries seem to be affected by the burst of internet speculative bubble from 2000 to 2001 causing structure change. Therefore, this study divides the research period into three periods of 1980-2018, 1980-2001 and 2002-2018 to obtain more robust results.
- This study focuses on the short-run Granger causality, whether let me add the ARDL approach to research the long-run cointegrated relationship in another paper in the future. The bootstrap panel Granger causality test can highlight the unit root (nonstationary) and cointegration characteristics of variables, that is, the verification process does not need the pre-tests of unit root and cointegration.
- The introduction and conclusions have been rewritten in this study.
- Attached is the revised full text.

Reviewer 3 Report
The abstract does not reflect the most important essence of the article. Chapters 1 and 2 are based on the literature on the subject, but this is a very poor literature review. There are no major conclusions, no critical discussion, no polemics. What were the selection criteria in Chapter 5 of these countries? - specifically. The conclusions are very poor, this is a brief summary. The literature on the subject should be supplemented with further publications.
Author Response
Thanks for your comments and suggestions.
Responses:
- The abstract, introduction, and conclusions have been rewritten in this study. Some papers are added in the literature review.
- This study focused on the top four Asian primary energy consumers, namely, China, India, Japan, and South Korea. The study period was from 1980 to 2018, covering 39 years of data. From the Global Energy Statistical Yearbook 2019, the energy consumption of these four countries in 2018 (in million metric tons of oil equivalent, Mtoe) is 3,273.47 in China (first in the world), 809.15 in India (third in the world), 454.14 in Japan (fifth in the world), and 301.02 in South Korea (eighth in the world).
- Some papers are added in the literature review.
- Attached is the revised full text.

Reviewer 4 Report
Dear Author,
It is very commendable that you approached such a topic, which can be of considerable interest for both academics and practicians.
For the literature review part I could recommend some new papers:
Human capital, energy and economic growth in China: evidence from multivariate nonlinear Granger causality tests:Fang, Z (Fang,Zheng); Wolski, M (Wolski, Marcin), EMPIRICAL ECONOMICS, 2019; The causal relationship among electricity consumption, economic growth and capital stock in Iran: (Salmanzadeh-Meydani, N., Ghomi, S. M. T. Fatemi) JOURNAL OF POLICY MODELING, 2019.
In the introduction you should mention the likely change in energy efficiency use across studied period and for sample countries, especially during the last 11 years since the advent of the global financial crisis. I do not think the energetic efficiency remained the same during the entire 39 years period! That means that if energy efficiency changed you can get economic growth with the same amount of energy spent or even with a lower amount of energy. This leads to the question: Why haven't you tried using a production function which correlates such inputs as energy with the output? Such as in the paper Energy and economic growth in developing Asian economies: Azam, Muhammad, JOURNAL OF THE ASIA PACIFIC ECONOMY, 2019.
I think you should divide the studied period into more homogeneous sub-periods to see whether your correlations and conclusions still hold or they change with the change of underlying economic and technological conditions and changes.
Also, you should exert more caution when making a statement. For example you mention that a lower oil price ”For Japan, it will help economic growth, the Japanese yen will depreciate more”. What is the connection between lower oil price and the depreciation of Yen? When industry becomes more competitive, it causes a depreciation in the currency?
The conclusions part needs definite improvement. You should mention how your study relates to other authors' findings (does it confirm, contradicts or finds new things...)
English needs considerable revision, starting with the first sentence..."Crude oil is an important human resource"...Human resource means a totally different thing...
After such a revision the paper could become eligible for publishing.
Author Response
Thanks for your comments and suggestions.
Responses:
- The introduction and conclusions have been rewritten in this study.
- Some papers suggested by you are added in the literature review.
- From Figure 3, it can be seen that the GDP and primary energy consumption (PEC) of these four countries seem to be affected by the burst of internet speculative bubble from 2000 to 2001 causing structure change. Therefore, this study divides the research period into three periods of 1980-2018, 1980-2001 and 2002-2018 to obtain more robust results.
- This study has deleted this sentence: “lower oil price and the depreciation of Yen”.
- Attached is the revised full text.

Reviewer 5 Report
This paper provides an interesting contribution to the energy economics focusing on the energy-growth nexus in Asia. However, the presentation of the manuscript can be improved. First, having professional proofreading of manuscript is desirable, especially the introduction. Second, all the variables listed in the equations must be appropriately defined. Third, the frequency of the data must be clearly stated and summary statistics for the variables should be presented and discussed. Finally, it would be worthwhile contrasting the results on the energy-growth nexus to those to the electricity-growth nexus, especially in Asia. See for example Chen & Chen (2007) and Fakih & Marrouch (2015) for a more recent overview of results from the energy economics literature.
References
Chen, S. T., Kuo, H. I., & Chen, C. C. (2007). The relationship between GDP and electricity consumption in 10 Asian countries. Energy policy, 35(4), 2611-2621.
Fakih, A., & Marrouch, W. (2015). The electricity consumption, employment and growth nexus: evidence from Lebanon. OPEC Energy Review, 39(3), 298-321.
Author Response
Thanks for your comments and suggestions.
Responses:
1.This study have finished the Academic English Language review.
2.This study have added descriptive statistics.
3.The introduction and conclusions have been rewritten in this study.
4.Some papers suggested by you are added in the literature review.
5.Attached is the revised full text.

Round 2
Reviewer 1 Report
Must be improved. You must use data in constant prices to real GDP, then you can see that conclusion will be different.
Author Response
Thanks for your comments and suggestions.
Responses:
This study first uses the bootstrap causality test to determine the relationship between energy consumption and real GDP for the top four major energy consumers in Asia, namely, China, India, Japan, and South Korea. This test can effectively overcome cross-sectional correlation and heterogeneity problems. Moreover, few studies investigate the relevance of real GDP to energy consumption, even though real GDP adjusted by inflation provides an accurate picture of a country’s economic situation. The results of this study will contribute to the academic and practical fields.
The real GDP data (in constant 2010 million USD) and nominal GDP data (in current million USD) are taken from the World Bank. Real GDP provides an accurate picture of a country’s economic situation because it can be easily compared to past data adjusted by inflation. Therefore, whether a country’s situation is better or worse year by year can be inferred.
During the 2002-2018 period, an independent causal relationship is observed between real GDP and PEC for all four countries, thus supporting the neutrality hypothesis. This study also compares the results of real GDP with those of nominal GDP. A unidirectional causal relationship is observed running from PEC to nominal GDP in South Korea, thus supporting the growth hypothesis that energy consumption has a significant effect on nominal GDP. A unidirectional causal relationship is observed running from nominal GDP to PEC in India, thus supporting the energy conservation hypothesis that nominal GDP has a significant impact on energy consumption. As mentioned above, the relationship between PEC and real GDP adjusted by the GDP deflator is weaker than that between PEC and nominal GDP.
Nominal GDP strengthens its relationship with PEC through the effect of prices for all the goods and services produced in an economy. The innovative findings of this study will contribute to this line of research.
Attached is the revised full text.

Reviewer 3 Report
Accept
Author Response
Thank you for accepting my manuscript. I will continue to make progress in article writing skills.
Reviewer 4 Report
You have observed some of my recommendations, that is dividing the period into more homogeneous subperiods and improving references.
With English you still have to work. You cannot use ”:” followed by a full sentence as below. ”Through 15 rigorous empirical research, we summarize the main conclusion as follows: The GDP and primary 16 energy consumption (PEC) of these four countries seem to be effected by internet speculative 17 bubble burst from 2000 to 2001”.
You repeat this in the conclusions part. So do not use ”:” followed by full sentences.
In the Conclusions part you made no references to similar studies and their results. You should refer to other authors' results and how you position your results.
I also think you can present a measure Energy/GDP which could show how much energy was needed to obtain 1 mil. $ of GDP. Its evolution would be interesting to follow.
Author Response
Thanks for your comments and suggestions.
Responses:
- “.” is being substituted for “:” by the end of full sentences.
- In the conclusions part of this study had made some references to similar studies and their results. For example, during the 1982-2018 period, an independent causal relationship is observed between real GDP and PEC for all four countries, thus supporting the neutrality hypothesis. This finding is similar to those of Fang and Wolski (2019) and Chiou-Wei et al. (2008), thus supporting the neutrality hypothesis in China and in the United States, Thailand, and South Korea, respectively. During the 1982-2001 period, a unidirectional causal relationship is observed running from PEC to real GDP in India, thus supporting the energy growth hypothesis that PEC has a significant impact on real GDP. This finding is similar to those of Azam (2019) and Chiou-Wei et al. (2008), thus supporting the energy growth hypothesis in developing Asian economies and in Taiwan, Hong Kong, Malaysia, and Indonesia, respectively.
- The study showed how much energy was needed to obtain 1 mil. $ of GDP. For example, from the coefficient analysis results of the independent variables, PEC in Table 10, we can see that the sum of the coefficients of lags 1 and 2 is -0.39109 for India. This finding means that one Mtoe increase in PEC will cause real GDP to decrease 0.391089 million USD in the following two years.
- Attached is the revised full text.

Round 3
Reviewer 1 Report
According to the data used and results, however, the conclusions in comparison of results and the discussion can be improved.